# Classification of Sleep Quality and Aging as a Function of Brain Complexity: A Multiband Non-Linear EEG Analysis

**DOI:** 10.3390/s24092811

**Published:** 2024-04-28

**Authors:** Lucía Penalba-Sánchez, Gabriel Silva, Mark Crook-Rumsey, Alexander Sumich, Pedro Miguel Rodrigues, Patrícia Oliveira-Silva, Ignacio Cifre

**Affiliations:** 1Facultat de Psicología, Ciències de l’Educació i de l’Esport (FPCEE), Blanquerna, Universitat Ramon Llull, 08022 Barcelona, Spain; lucia.penalbasanchez@med.ovgu.de (L.P.-S.);; 2Human Neurobehavioral Laboratory (HNL), Research Centre for Human Development (CEDH), Faculty of Education and Psychology, Universidade Católica Portuguesa, 4169-005 Porto, Portugal; 3Department of Psychology, Nottingham Trent University (NTU), Nottingham NG1 4FQ, UK; 4Institute of Cognitive Neurology and Dementia Research (IKND), Otto-von-Guericke-University Magdeburg (OVGU), 39120 Magdeburg, Germany; 5Centro de Biotecnologia e Química Fina (CBQF)—Laboratório Associado, Escola Superior de Biotecnologia, Universidade Católica Portuguesa, 4169-005 Porto, Portugal; 6UK Dementia Research Institute (UK DRI), Centre for Care Research and Technology, Imperial College London, London W1T 7NF, UK; 7UK Dementia Research Institute (UK DRI), Department of Basic and Clinical Neuroscience, Maurice Wohl Clinical Neuroscience Institute, London SE5 9RX, UK

**Keywords:** sleep quality, PSQI, EEG, non-linear multiband analysis, brain complexity, classification, machine learning, healthy aging

## Abstract

Understanding and classifying brain states as a function of sleep quality and age has important implications for developing lifestyle-based interventions involving sleep hygiene. Current studies use an algorithm that captures non-linear features of brain complexity to differentiate awake electroencephalography (EEG) states, as a function of age and sleep quality. Fifty-eight participants were assessed using the Pittsburgh Sleep Quality Inventory (PSQI) and awake resting state EEG. Groups were formed based on age and sleep quality (younger adults *n* = 24, mean age = 24.7 years, *SD* = 3.43, good sleepers *n* = 11; older adults *n* = 34, mean age = 72.87; *SD* = 4.18, good sleepers *n* = 9). Ten non-linear features were extracted from multiband EEG analysis to feed several classifiers followed by a leave-one-out cross-validation. Brain state complexity accurately predicted (i) age in good sleepers, with 75% mean accuracy (across all channels) for lower frequencies (alpha, theta, and delta) and 95% accuracy at specific channels (temporal, parietal); and (ii) sleep quality in older groups with moderate accuracy (70 and 72%) across sub-bands with some regions showing greater differences. It also differentiated younger good sleepers from older poor sleepers with 85% mean accuracy across all sub-bands, and 92% at specific channels. Lower accuracy levels (<50%) were achieved in predicting sleep quality in younger adults. The algorithm discriminated older vs. younger groups excellently and could be used to explore intragroup differences in older adults to predict sleep intervention efficiency depending on their brain complexity.

## 1. Introduction

Good sleep quality is essential for maintaining one’s cognitive and mental health over their lifespan [1,2,3], with critical repercussions at a personal, societal, and economic level. Understanding poor sleep quality in older adults is particularly important, as it is associated with the risk of several non-communicable diseases (such as diabetes, cardiovascular diseases, and obesity) and may contribute to cognitive decline and memory impairments [4,5,6,7] through mechanisms such as increases in blood pressure, evening cortisol levels, proinflammatory cytokines, and sympathetic tone [6,7,8].

Changes in sleep patterns are part of a healthy aging process [9], but these changes do not imply a decrease in sleep quality. Research shows that there are many older individuals who report having optimal sleep quality or even better sleep quality than they had in middle age and that they remain free of sleep disorders such as obstructive sleep apnea or insomnia [3,10]. Age-related changes such as reduced brain electrical activity and synchronization [11], minor decreases in white matter volume, and cortical atrophy may contribute to cognitive decline and consequently to poor sleep quality [11,12]. Also, a poor sleep quality might exacerbate neurocognitive changes that occur as a function of age. 

This bidirectional relationship between cognition and sleep quality in the context of aging might explain the reason why sleep quality interventions have a stronger effect in younger adults in comparison to older adults, especially in cognitive outcomes [3]. Older adults included in these studies might present higher rates of atrophy in the hippocampus and frontal regions, areas involved in the acquisition and consolidation of new memories. These participants, despite presenting an improvement in physiological sleep such as an increase in deep sleep waves and sleep spindles, could not strengthen memories due to reduced thalamic–cortical network support [3]. The increased physiological sleep quality might be beneficial for slowing down the cognitive decline both in aging and neurodegenerative diseases long-term. In other words, brain atrophy and cognitive decline present at the baseline, (i.e., start of the interventions) cannot be reversed. However, the pace of further atrophy and cognitive impairment might be slowed down if patients incorporate these interventions into their daily routine as quality physiological sleep facilitates the removal of pathological proteins, notably beta-amyloid and tau, from the brain, making the individuals more resistant to them [3,13,14].

A concept intimately linked with brain resistance is brain complexity, defined as the ability of the neuronal circuits to interact at different spatial and temporal scales, enabling the individual to flexibly adapt to the environment [15]. Brain complexity has been associated with increased health and a greater probability of survival, and is impacted by sleep quality and decreased in the presence of cognitive impairment and neurodegenerative diseases [15,16]. Healthy aging has been associated with a shift in the local/global complexity balance, with more information being encoded at a local level and less at a global one [17]. Absence of this shift in older individuals predicts worse cognitive outcomes. One of the functions of sleep is regulating the complex organization of the dynamic brain by balancing the cortical excitatory–inhibitory activity [18]. Classification of individuals by brain complexity may aid personalized interventions for sleep issues, integrating awake resting state brain function, sleep questionnaires, non-linear signal processing, and machine learning for efficiency [1].

Electroencephalography (EEG) is a widely used tool in sleep medicine that measures a gross signal from extracellular, post-synaptic potentials from thousands of neurons, primarily pyramidal cells [19]. It has an excellent temporal resolution and is inexpensive and easy to transport [20]. Most studies exploring sleep quality and/or aging with waking EEG data have used linear methods of analysis, treating the whole time series as deterministic (e.g., computing the mean of the whole timeseries). However, under deterministic chaos, the brain is neither stochastic nor completely predictable [21]. Considering this, the theory of non-linear dynamics and chaos has been proposed as a better way for biophysiological data analysis [22,23,24]. Under this paradigm, non-linear features extracted from EEG time series are utilized to explore various aspects of the human brain, including the interplay between brain complexity, age, and sleep [25,26].

Recently, descriptors of complexity have been extracted from EEG signals to understand aging processes and sleep quality [25]. The most used are the Correlation Dimension (D2), Long-Range Temporal Correlations (LRTCs), energy, and entropy. The following paragraphs are dedicated to defining these measures and presenting studies that used them in the context of aging or sleep quality.

D2 is a measure of connectedness of the system, and is reduced following sleep deprivation, reflecting a decrease in topological complexity [27].

LRTC measures the memory of a system (i.e., how slowly the autocorrelation of a timeseries decays). One of the most common ways to extract the LRTC is by computing the Hurst Exponent (H), which is a measure of statistical self-dependence of the brain activity over multiple scales of time and space. H provides information about the self-similarity of the time series. Colombo et al. (2016) [18] used this exponent (H) and showed that participants with worse sleep quality presented a higher LRTC, suggesting a decrease in brain-balanced excitability which is translated into a decreased ability of the cells to regulate its activity.

Energy measures signal power over time, i.e., strength or intensity of EEG signals across different frequency bands [26]. A comparison of several machine learning methods—support vector machine (SVM), k-nearest neighbor (KNN), and discriminative graph regularized extreme learning machine (GELM)—supported GELM as the optimal classifier in discriminating young adults with varying amounts of sleep quantity, based on gamma frequency (62.16% accuracy, 83.57% when selecting optimal channels) [26]. 

Entropy is a measure of complexity that indicates the level of disorder in a dynamic system. High levels of entropy signify less order and increased irregularity, which translates into a dysfunctional brain organization (which may require compensation by other networks). Increased entropy is seen in healthy aging [28]. Entropy studies also suggest reduced complexity with age and less connectedness across hemispheres and modules [29]. These brain features typically detected in the aging brain possibly reflect a reduced repertoire of behaviors or a reduced flexibility to adjust to different situations [29]. Recently, healthy older adults and older adults with mild cognitive impairment (MCI) were more accurately classified by combining measures of complexity into a single algorithm, supporting non-linear measures of complexity in understanding healthy and unhealthy aging [30]. 

Until now, previous studies that investigate the effect of both age and sleep quality have used (1) one non-linear feature and classical statistics to explore differences in sleep between groups of young adults [18,27,31]; (2) one or more non-linear features (i.e., when using more than one these were explored separately) and classical statistics to explore differences in brain configuration between young and older adults [23]; (3) one or more non-linear features and ML techniques to classify sleep [27]; and (4) one non-linear feature and ML to classify younger adults vs. older adults [29]. 

The overall aim of the present study was to determine whether age and self-reported sleep quality impact the complexity and stability of the brain detected in wake EEG. More specifically, it aimed to classify individuals by their range of age (i.e., younger vs. older adults) and their sleep quality (i.e., good vs. bad). To do so, a combination of novel algorithms that combine 10 EEG non-linear features of complexity, namely energy, activity, self-similarity, and connectedness, were used. This study diverges from previous research that predominantly utilized linear EEG analyses by adopting a non-linear dynamic approach to EEG data. By incorporating multiband decomposition and extracting distinct non-linear features, the methodology of this study intends to provide a better understanding of brain complexity. This advanced analysis enhances our insight into how brain activity varies with sleep quality and age changes, offering a more comprehensive view than traditional linear methods. Additionally, by integrating these features into a robust classification system, we provide a methodological advancement that can more accurately reflect the real-world complexity of brain behavior interactions.

Our questions and expected outcomes are articulated as follows: Can the proposed algorithm effectively categorize individuals’ brain complexity based on age? (e.g., distinguishing between younger individuals with good sleep quality and older individuals with good sleep quality). Previous studies have noted an increase in slower frequencies among older adults [32]. Additionally, some researchers have suggested a reduction in complexity at a global network level and an increase at a local level [17]. Therefore, we hypothesized that the algorithm would successfully classify participants by age, yielding higher accuracy rates, particularly in the theta and delta sub-bands.Is the proposed algorithm capable of better distinguishing between older and younger individuals when one group experiences compromised sleep quality? The extent to which participants can be classified based on both sleep quality and age remains uncertain. However, considering the observed alterations in complexity associated with sleep and aging, higher accuracies in distinguishing between young and older individuals were anticipated, particularly when comparing young adults with good sleep quality to older adults experiencing sleep disturbances. Essentially, we expected these two groups to exhibit the greatest dissimilarities, as we incorporate variations in sleep quality alongside the aging process.Does sleep quality affect the awake resting state brain complexity and stability in young adults and in older adults? Which sub-bands and regions enable a better classification level? (YG vs. YB and OG vs. OB). As mentioned previously, sleep quality is associated with a decrease in complexity [33], hence discrimination between these pairs of groups is expected although with lower accuracy levels than when contrasting groups of different ages.

## 2. Materials and Methods

### 2.1. Participants

The cohort used in this study overlaps with that used in a previous publication [34]. The study was approved by the Health Research Authority, UK (REC reference: 17/EM/1010), and participants provided informed consent. 

A total of 58 right-handed volunteers were included in the current study. Participants were divided into four groups (see below) depending on their age (i.e., younger adults aged 20–34, older adults aged ≥ 65) and their sleep quality (good vs. bad), as assessed using the PSQI [35]. For a PSQI description, see section “data-description”.

Group 1: *n* = 11 young adults with good sleep quality (YG) (scores < 5 in PSQI); *n* = 5 females and *n* = 6 males (mean age = 23.36; *SD* = 2.70).

Group 2: *n* = 13 young adults with bad sleep quality (YB) (scores > 5); *n* = 5 females and *n* = 8 males; (mean age = 25.53; *SD* = 3.54).

Group 3: *n* = 9 older adults with good sleep quality (OG) (scores < 5 in PSQI [35]); *n* = 4 females and *n* = 5 males; (mean age = 73.77; *SD* = 5.45).

Group 4: *n* = 25 older adults with bad sleep quality (OB) (scores > 5 in the PSQI); *n* = 17 females and *n* = 8 males; (mean age = 72.56; *SD* = 3.40).

Young participants were recruited from the Nottinghamshire area. Older participants were recruited through the Trent aging panel, an internal Nottingham Trent University database of older adult study volunteers.

Participants presented normal or corrected-to-normal vision; no history of psychiatric, cognitive, or neurological disorder; and no medication that could interfere with the EEG recordings. Participants were asked to not consume alcohol 24 h before the recordings and caffeine and nicotine 3 h prior. To guarantee that none of the participants presented MCI, they were assessed with the Hopkins Verbal Learning Test-Revised (HVLT-R) [36]. The HVLT-R in comparison to other scales such as the Mini-Mental State Examination (MMSE) [37] has very high sensitivity and specificity enabling it to capture subtle differences in cognitive decline. 

### 2.2. Data Description

#### 2.2.1. Sleep Quality Assessment (PSQI)

Sleep quality was assessed using the standardized self-rated questionnaire PSQI [12,35]. It is composed of 19 items that measure 7 domains: sleep quality, sleep latency, sleep duration, sleep efficiency, sleep disturbances, use of sleep medication, and daytime dysfunction. All domains are summarized in a global score. Given poor sleep quality in normal aging has been associated with the elements measured by the PSQI [35], the global PSQI score was used to differentiate bad vs. good sleep in the current study. This tool has been reported to be optimal for assessing sleep quality. It has a diagnostic sensitivity of 89.6 and a specificity of 86.5 (kappa 0.75 *p* < 0.001). Scores range from 0 to 21. Scores > 5 are indicative of poor sleep or significant sleep disturbance [35].

#### 2.2.2. EEG Data Collection

Eyes closed resting state EEG data was recorded using a 128-channel Active Two Acquisition system (BioSemi, Amsterdam, Netherlands) at a sampling rate of 2048 Hz and processed at 24-bits. Seven additional channels were applied around the face to help with artifact detection. To reduce the computational load of data, 32 channels were used for the data analysis: ‘A1’, ‘A7’, ’A15’, ‘A17’, ‘A19’ ‘A23’, ‘A28’, ‘A30’, ‘B2’, ‘B4’, ‘B11’, ‘B16’, ‘B22’, ‘B26’, ‘B29’, ‘C4’, ‘C7’, ‘C11’, ‘C15’, ‘C16’, ‘C21’, ‘C24’, ‘C28’, ‘C29’, ‘C30’, ‘D4’, ‘D10’, ‘D16’, ‘D19’, ‘D23’, ‘D26’, ‘D31’. The location of channels captures the electrical activity through all the regions of the scalp. The use of this scalp widespread 32 channels has been reported to be optimal by several researchers [19].

### 2.3. EEG Data Preprocessing

MATLAB ver. R2018a and EEGLAB [38] were used to preprocess the EEG data. Raw EEG data was converted into 32 real value data vectors representing data extracted from each of the EEG channels. Data was imported and referenced to linked mastoids, high and low pass filtered between 0.1 Hz and 45 Hz, downsampled to 256 Hz and the DC component has been removed. Bad channels were visually inspected, manually removed, and interpolated. An independent component analysis (ICA—runica) was used to discard those components that showed ocular and muscular artifacts (i.e., runica: visual inspection of scalp topographies; and activity spectra: rejection of noisy data “eye blinking/muscle”). To reduce computational demands, from the five minutes of EEG recorded data, only one minute was used for the data analysis (i.e., a total of 15,361 time points and 32 channels). The first minute of the signal was selected to capture the most attentive moment and avoid sleepiness in older adults with bad sleep quality. Revising other studies shows that one minute of data is sufficient to conduct this type of analysis [39]. 

### 2.4. EEG Signal Processing and Feature Extraction

The subsections below describe the steps followed by the data pre-processing. These include multi-band decomposition, feature extraction, data normalization, and classification (Figure 1).

#### 2.4.1. Multiband EEG Decomposition

For each participant and channel, the EEG time series were split into 5 s windows (i.e., a total of 12 windows). After, from each 5 s widowing analysis per channel an EEG signal decomposition into frequency, sub-bands were performed for each participant, per channel. EEG sub-bands delta (δ, 0.1–4 Hz), theta (θ, 4–8 Hz), alpha (α, 8–16 Hz), beta (β, 16–32 Hz), and gamma (γ, 32–45) were extracted from the broadband signal using discrete wavelet transform (DWT). 

DWT is one of the most used tools to perform time-frequency analysis for non-stationary data [40]. In contrast to the classic Fourier Transform, where the frequency is extracted but time frequency is lost, using DWT, wavelets are localized both in frequency and time, ensuring optimal time and frequency resolutions.

DWT was used in this study, with a biorthogonal 3.5 wavelet. This type of wavelet is often preferred as it adjusts to the EEG’s original signals with very little deformation [41]. DWT was performed through an octave band critically decimated filter bank [42,43]. The signal was transformed into approximation and details, using a scalar function and a wavelet function. The values obtained for each participant, channel, and sub-band were used as input values for the feature extraction. 

#### 2.4.2. EEG Non-Linear Analysis 

The non-linear nature of the EEG data was assessed with the tool provided by [44]. Then, two main steps were conducted: (1) reconstruction of the attractor from the state space from observations and (2) extraction of features of EEG complexity and variability; (2.1) from attractor (from the state space): correlation dimension, Lyapunov exponent and approximate entropy (descriptors of the attractor); and (2.2) from time series: long-term memory measures, fractal measures, energy, and entropy. 

Attractor reconstruction from the state space with time delay embedding:

Detecting an order or structure behind the EEG time series is a challenging task as the data is complex and chaotic [45]. In consequence, the extraction of certain descriptors or features directly from the time series is not an easy procedure. However, when reconstructing this type of data from “time series” to “state space” a hidden order can be observed. The state space represents every single state of the dynamic system, the brain, in an m-dimensional plot forming a geometric structure called an attractor. Note that the state of a dynamic system can be defined as the configuration of the system at a specific time. In the present study, each state is represented by EEG channel values at a specific time point of the time series. The most implemented approach to reconstructing the phase space of EEG signals is the “time delay embedding”. The minimal dimension of the state space (acquired by “embedding”) enables the extraction of non-linear features to explore the whole dynamic system and the interactions within it in a non-ambiguous way. Some of these non-linear features are the topology (connectedness), general structure, prediction of states, correlation dimension, and causality between variables [30,46]. 

In the present study, a reconstruction of the state space using time delay embedding is given by:
*x_i_* = [*x*(*i*), *x* (*i* +τ), …, *x*(*i* + (*m* − 1) τ)],(1)
where τ is the incorporation delay and *m* is the dimensionality. The values τ and *m* were obtained following the methods in Faust and Bairy (2012) [47]. The vector sequence$\;x_{i}$, *i* = 1, 2, …, *M*, where, *M* = *N* − (*m* − 1)τ, form the reconstructed attractor [30,47]. 

### 2.5. Feature Extraction

The 10 features presented below were extracted per channel (32 channels), per each participant, and per each sub-band.

#### 2.5.1. Features Extracted from Reconstructed Attractor

Once the phase space was determined, the correlation dimension, Lyapunov exponent, and approximate entropy were extracted. These measures enable us to determine the complexity and balance of the brain:

Correlation dimension (D2) is a measure that describes the complexity of the system based on the topology or connectedness of the attractor. In other words, it estimates the space and distribution occupied by different points of the fractal attractor. For instance, two points in the attractor might be very close in time but far in space. It is estimated based on the correlation integral, a function of variable distances:(2)$$C\left(r,\;M\right)=\frac{2}{M\left(M-1\right)}\sum_{i=1}^{M}\sum_{J=1;\;J\;\neq \;i}^{M}\theta \;(r-||x_{i}-x_{j}||)$$
where *M* is the number of data points or length or the attractor and Θ is the Heaviside function meaning that this function attributes a value of 0 for negative inputs and 1 for positive ones. *C*(*r*) determines the probability that two pairs of points of the attractor {*x*_i_, *x*_j_} present a distance between them equal to or less than *r* [30,46,47]. From this, the correlation dimension can be estimated as:(3)$$D_{2}=\underset{r\to 0}{lim}\;\frac{logC(r,\;M)}{log(r)}$$

Lyapunov exponent (LLE) measures the stability of the attractor and quantifies chaos. Chaotic or strange attractors perform two processes: (1) a process of expansion that consists of trajectories starting from the same or similar point diverging and (2) a process of folding as time evolves, in other words, trajectories go back to the initial state converging (close to each other). LLE determines the rate of expansion and folding. The larger the rate (LLE), the more chaotic the attractor. The LLE rate of an attractor should be a positive value to be chaotic. For each state of the state, the largest exponent LLE can be extracted by finding the state *x*_j_ that satisfies *min*_j_ ||*x*_i_ − *x*_j_||. The estimates are given by [30,48].
(4)$$\lambda \left(i\right)=\frac{1}{M+2}\sum_{K=1}^{M}\frac{1}{kTs}\;\mathrm{In}\;\frac{\left||x_{i+k}-x_{j+k}|\right|}{\left||x_{i}-x_{j}|\right|}$$
where *T*_s_ is the sampling period. The LLE is defined by the slope of the best linear approximation of *λ*(*i*) [48].

Approximate entropy (ApET) computes the rate at which information of the dynamic system is lost over time [30]. It is defined as:(5)$$\mathrm{ApET}\left(m,r,N\right)=\frac{1}{N-m+1}\sum_{i=1}^{N-m+1}log[C_{i\;}^{m}\left(r\right)]-\frac{1}{N-m\;}\;\sum_{i=1}^{N-m}\log\left[C_{i}^{m+1\;}\left(r\right)\right],$$
where
(6)$$C_{i\;\;\;}^{m}\left(r\right)=\frac{1}{N-m+1}\sum_{j=1}^{N-m+1}\theta (r-\left||x_{i}-x_{j}|\right|)$$
is the probability of the point $x_{i}$ on the attractor to be segregated from the other points by a distance inferior or equal to *r*. 

#### 2.5.2. Features Extracted Directly from the Time Series

In this subsection, the features extracted directly from the time series are described. These are long-term memory measures (Hurst Exponent, Detrended Fluctuation Analysis), fractal dimension measures (Higuchi and Katz Algorithm), energy, and entropy.

##### Long Term Memory Measures

The Hurst Exponent (*H*) is used to assess long-range statistical self-dependence of a time series (i.e., self-correlation, smoothness, and self-similarity of a single time series) [30,47]. It can be estimated as:(7)$$H=\frac{log(R/SD)}{log(N)},$$
where *R* is the range (maximum–minimum inside the series) and *SD* the standard deviation. *H* is estimated by the slope of the best linear approximation of log[*R*(*n*)/*SD n*)] as a function of log(*N*), see [26] for computation details of *R*(*n*)/*SD* (*n*). The more irregular the EEG signal is, the closer to 0 *H* will be [46].

Detrended Fluctuation Analysis (Δ) is similar to the Hurst Exponent in that it measures the statistical dependency on non-linear signals. However, this latter one explores exclusively self-similarity, in other words, long-range correlations of a time series [30,49,50]. From *x*(*n*), the cumulative deviation series is calculated as follows:(8)$$y\left(k\right)=\sum_{i=1}^{k}\left[x\left(i\right)-\bar{X}\right].$$

A linear approximation denoted by *y_m_*(*k*) is estimated for each m-long segment of y(k). The following formula defines the signal’s average fluctuation as a function of m [47]:(9)$$F\left(m\right)=\sqrt{\frac{1}{N}}\sum_{k=1}^{N}{\left[y\left(k\right)-y_{m}\left(k\right)\right]}^{2}.$$

The scale exponent Δ signifies the correlation properties of the signal *x*(*n*), represented by the slope of the best linear approximation of log *F*(*m*) as a function of log *m* [50]. 

#### Fractal Dimension Measures

Fractal Dimension with Higuchi algorithm (FDh): a fractal is a geometric figure that is divided by smaller identical subfigures, and presents self-similarity at different scales. This type of figure is used to model and assess real-world problems as its shape is more natural than conventional geometric figures. The brain presents attractors with the structure of a fractal. In EEG processing, the fractal dimension measures the complexity of the brain by detecting transient events in the waveforms [47]. This feature can be calculated directly from the signals, meaning that reconstruction of the attractor is not needed. There are several algorithms to compute the FD. In the present study, the Higuchi Algorithm was used due to its excellent capacity for accuracy achieved in seminal research [47].

For *m* = 1, …, *n* and *k* = 1, …, *k*_max_, where *k*_max_ is obtained experimentally despite *k*_max_ = 8 being initially proposed, a distance measure is computed as [30,46,47].
(10)$$L_{m}\left(K\right)=\frac{N-1}{\left[a\right]k}\sum_{i=1}^{\left[a\right]}|x(m+ik)-x(m+(i-1)k|,$$
where *a* = (*N* − *m*)/*k* and ⌊*a*⌋ represents the largest integer equal to or less than a. The average distance is computed as $L(k)=\sum_{m}^{k}=L_{m(k)/k}$ for *k* = 1, …, *k*_max_. The FD estimate, denoted by FDH, is then given by the slope of the best linear approximation of *ln*[*L*(*k*)] as a function of *ln*(1/*k*).

Fractal dimension with Katz Algorithm (*FD_K_*): additionally, the Katz [51] algorithm (*FD_K_*) was used to determine FD:(11)$${FD}_{K}=\frac{log(L/a)}{log(d/a)},$$
where *L* is the sum of the distances between the successive points of *x*(*n*), a is the average distance between the successive points, and d is the greatest distance between *x*(1) and the remaining points of *x*(*n*).

#### Energy and Entropy

Energy (*EN*): energy is one of the most used measures for explore aging processes. It detects the slowing down of brain frequencies or the shifts from high frequencies to low frequencies in aging [30].
(12)$$EN=\sum_{n=1}^{N}{\left|x\left(n\right)\right|}^{2}.$$

Entropy (ETs and ETL): similarly to LLE, entropy measures the loss of information of its dynamics. A positive entropy denotes chaos, meaning that it takes more time to expand than to fold back and produces more information than it destructs. Entropy detects the amount of randomness or uncertainty in the EEG signal. In other words, it assesses how ordered or disordered the peaks of the signal are. A low entropy reflects predictability or repetition in the EEG signal patterns. The Shannon (ETs) and Logarithmic (ETL) entropies [30,52,53] can be estimated as:(13)$${ET}_{S}=-\sum_{n=1}^{N}{\left|x\left(n\right)\right|}^{2}\log\left[{\left|x\left(n\right)\right|}^{2}\right]$$
and
(14)$${ET}_{L}=-\sum_{n=1}^{N}\log\left[{\left|x\left(n\right)\right|}^{2}\right]$$

After extracting the non-linear features DE, LLE, H, Δ, FDh, FDk, EN, ETs, ETL, and Apet per each sub-band and window over channels, the time series vectors were compressed by mean stat over channel and sub-band and then normalized using z-score per each pair of groups, channels, and sub-bands (i.e., YG and OB, YB and OB, YG and OG, OB and OG, YG and YB). The normalized values obtained were utilized as input for a variety of machine learning techniques, including Support Vector Machine (SVM), K-Nearest Neighbors (KNN), Logistic Regression (Log.reg), and decision trees. These techniques were applied based on pairs of groups, channels, and sub-band organization, facilitating comprehensive analysis along channels and comparison across different configurations and deeper analyses per sub-bands along different channels. This approach allowed for a thorough exploration of data patterns and relationships, leveraging the strengths of each machine learning algorithm to uncover insights.

To ensure the results are generalizable, a leave-one-out cross-validation procedure was used. Due to the limited amount of data, all were used in the cross-validation. Table 1 indicated the used classifiers and the optimized hyper-parameters using by default the Matlab Classification Learner APP pre-designed models.

## 3. Results

### 3.1. Tomographic Maps for Discrimination over Scalp

The topographic maps in Figure 2 display the results of the best classifier for each pair of groups. Mean accuracies were assessed to select the best classifier. See Appendix A where the accuracies reached in each classifier are displayed.

### 3.2. Discriminatory Capability of Used Classifiers

Generally, as displayed in Figure 3, the results showed an excellent mean accuracy capacity of the algorithms to discriminate the following groups [1] YG vs. OB (different age, only bad sleep in older adults) [2] YB vs. OB (different age, bad sleep in both), [3] YG vs. OG (different age, same good sleep), and [4] YB vs. OG (different age, bad sleep in young adults) (see points 1–4 in Figure 3 showing accuracy levels); a good accuracy [5] OG vs. OB (same age, different sleep quality) and a low accuracy [6] YG vs. YB (same age, different sleep quality). Considering these results, when comparing groups of different ages (i.e., YG-OB, YB-OB, YG-OG, YB-OG) “bad sleep” in combination with “older age” constitute the variables that allow better to differentiate the groups, followed by “aging” (independently of sleep) and last, “bad sleep” in young adults. Additionally, when comparing groups of the same age (i.e., OG vs. OB and YG vs. YB), the older groups are easier to discriminate against than the younger ones. This could be explained by the fact that brain complexity and energy are not that affected in young participants who do not sleep well or because the algorithms used are better at capturing differences between young and old.

Regarding mean accuracies in the different sub-bands, results showed an excellent classification accuracy in the alpha, theta, and delta sub-bands when comparing young vs. older adults, especially when comparing the YG vs. OB, with an accuracy of 80% in alpha, 82% in theta, and 85% in delta (Table 2). The older groups’ (i.e., OG vs. OB) mean discriminatory capacity in all sub-bands was lower than when comparing Y vs. O but preserved (over 70%). Lower mean accuracy levels were found when comparing the YG vs. YB (Table 2). Nonetheless, the classification accuracies of specific EEG channels show an optimal discrimination between the young groups in all the sub-bands except in gamma (see Table 2 YG vs. YB, highlighted in blue).

### 3.3. Differences in Specific Regions across Groups

In this subsection, the most relevant results of accuracy levels in specific channels will be presented.

(i)Young versus older adults

As displayed in the topography maps in Figure 2, YG vs. OB are the study groups that present a higher discriminatory capacity. Generally, the areas that show higher differences between age groups are the frontotemporal regions (affected in gamma, alpha, and theta). Additionally, some occipital and parietal regions are markedly affected, especially in the theta and delta sub-bands.

Regarding specific channels, the regions that enable a higher discriminatory capacity between the Y vs. O in the slow rhythms (i.e., delta and theta) are within the temporal-parietal and occipital (especially channel B16). This might indicate age-related changes independently of sleep quality, as it is present when comparing all pairs of groups Y vs. O, but is not present when comparing OG vs. OB nor YG vs. YB. Conversely, the occipital (EEG channel A30) seems to be related to bad sleep quality in O (see Table 2). This area cannot discriminate between young with good sleep vs. old with good sleep. Additionally, channel A23 in the occipital seems to be related to bad sleep both in young and older adults.

Regarding the alpha sub-band, the frontotemporal region C11 is the most different when comparing the YG vs. OB and is associated with a bad sleep quality only in older adults, as this result is also present when comparing the YB vs. OB and the OG vs. OB.

Results in the gamma and beta sub-bands in Y vs. O suggest that some occipital regions, especially the A23, are associated with bad sleep quality in older participants (this is evident as in all pairs of groups where OB is present, this region shows a great accuracy performance). Conversely, the left frontotemporal region D23 is associated with older adults with good sleep, as we can see only this region is different between YG-OG, and YB-OG (see Appendix A).

(ii)Old good sleep versus old bad sleep

When comparing the OG vs. OB, an overall ≥71% accuracy in all sub-bands was achieved. As mentioned in the last section, results show that regions C11 (frontotemporal in alpha) and A23 (occipital in gamma and beta) are associated with bad sleep in the older group.

(iii)Young good sleep versus young bad sleep

The highest mean accuracy achieved by the YG vs. YB was 50% in the alpha sub-band. However, specific channels in beta, alpha, theta, and delta reached accuracy levels > 70%.

The parietal central line (channel A19) in the alpha sub-band seems to be associated with bad sleep in young adults (note that the algorithms can discriminate this region between the groups YG vs. YB as well as between YB vs. OG). Other channels that allow differentiating between YG and YB but no other pairs of groups are channel D19 (parietal) in the beta, B29 (fronto-central) in theta, and D31 (inferior parietal) in the delta sub-band. This section may be divided by subheadings. It should provide a concise and precise description of the experimental results, their interpretation, as well as the experimental conclusions that can be drawn.

## 4. Discussion

The aim of this study was to classify young and older adults with good and bad sleep quality as a function of brain complexity. Whilst similar algorithms have been used to discriminate healthy older adults from those with neurodegenerative diseases using resting state EEG data [30], the current study is the first to demonstrate the utility of classifying groups based on healthy aging and sleep quality.

The algorithm achieved excellent mean accuracies when comparing young vs. older adults. Moderate–high accuracies exist when comparing the older adult groups (e.g., older with good sleep quality with older adults with bad sleep quality) and low accuracies when classifying the younger groups (e.g., younger adults with bad sleep vs. younger adults with good sleep quality). Additionally, the algorithm enabled excellent discrimination between all pairs of groups in specific sub-bands and regions. Our data showed that poor sleep quality, as indicated by higher PSQI scores, correlates with distinct alterations in EEG signals, particularly in frequency distribution and non-linear dynamics. This association suggests that sleep quality directly influences the complexity and stability of brain activity, which could affect cognitive and neurological health. For instance, our findings that bad sleep quality in older adults manifested differently in EEG patterns than younger adults with similar sleep scores highlight the interplay between aging and sleep quality on brain function.

Brain configuration in the lower frequencies such as alpha, theta, and delta sub-bands seems to play an important role in the aging process, especially in temporal and parietal regions. This can be seen when comparing the older vs. younger groups, independently of their sleep quality. This is aligned with several studies that indicate changes in older adults in the slower rhythms in several regions, achieving accuracy levels of 75.5% [54,55]. These changes have been hypothesized by several authors to be caused by a generalized slowing of the nervous tissue, a decrease in cerebral perfusion and metabolism and inhibition mechanisms [56].

Additionally, results showed that, although age allowed discrimination between groups (e.g., YG vs. OG), even a higher accuracy is achieved when both age and sleep quality are considered (i.e., young adults with good sleep quality vs. older adults with bad sleep quality). This may reflect an interaction between aging and sleep quality on brain function.

Evidence suggests that changes in theta and delta frequencies are associated with sleep deprivation and bad sleep quality [57]. According to a systematic review, the hypothesis underlying this finding is that delta is a marker of homeostatic sleep drive, and the longer we are awake or sleep deprived, the higher the delta [58]. Changes in the delta sub-band apart from denoting typical brain aging seem to be associated with sleep deprivation and bad sleep quality independent of age. Münch et al. [57] found that young cohorts show a more pronounced delta activity in frontal regions, while the older present a decrease during wakefulness. This, according to the authors, might indicate a “pre-frontal tiredness” due to bad sleep quality and aggregated “frontal tiredness” due to aging. Although our findings cannot determine directionality, they support alteration in frontal and temporal delta and theta sub-bands in older adults with bad sleep quality in comparison to younger adults with good sleep. Furthermore, our findings suggest that the occipital region is generally affected both in young and older adults with bad sleep quality. However, while the former group only presents alterations in the delta and theta sub-bands, the latter presents differences in the occipital across all sub-bands (especially in the gamma, beta, and alpha). This discovery is intriguing, especially considering that previous research on sleep deprivation has indicated a slower dominant occipital frequency rhythm in wake EEG within the occipital region, a characteristic more prevalent in older adults compared to younger adults [59]. One plausible explanation could be attributed to the nature of our algorithm, which not only extracts sub-bands but also accounts for the intricacies of brain complexity. However, further investigation is warranted to validate and replicate our findings.

When comparing young cohorts, the algorithm achieved good discrimination accuracies only in specific regions and sub-bands. More precisely, occipital and parietal regions seem to be affected in young adults with bad sleep quality in the alpha, beta, and delta frequencies. Additionally, left frontotemporal gamma and beta are associated with younger adults with good sleep.

The low mean classification accuracy of the young groups such as YG versus YB might reflect the fact that sleep does not significantly alter the brain configuration in younger adults and that bad sleep quality in older adults affects the brain configuration in a more widespread manner. Alternatively, the features and algorithms currently used may not be suited to discriminate between good and bad sleepers in younger groups.

This study presents some limitations. First, only the first minute of data was used to decrease processing computational time. Although it has been demonstrated that one minute of EEG data is sufficient for this type of analysis, the restriction to such a brief snapshot may overlook nuances that a more extended time series could reveal. Consequently, it is recommended that future studies extend the duration of EEG data analysed to determine whether the accuracy levels vary over different time windows. Additionally, assessing more extended time series could provide deeper understanding into the stability and variability of EEG features, potentially leading to more robust and generalizable findings. Second, a search for the best combination of non-linear features was not performed. A recent study conducted by some of the co-authors indicated that this combination of features is optimal for classifying healthy older adults and older adults with neurodegenerative diseases [30]. Accuracy levels in the younger groups might be improved by using another combination of features. This takes several days of computational work. We propose future studies to investigate whether another combination of features allows better discrimination of young samples. Third, the effects of age and the ones on sleep quality are not easy to disentangle and, although we can determine the classification accuracies with high confidence and which sub-bands are more affected in each pair of groups, answering the questions “which of these changes are more associated with a bad sleep quality and which ones with age?” or “is there a decrease or increase in delta or theta sub-bands”? is not possible. Only inferences can be made by comparing all the results of all pairs of groups. Fourth, this study has a small sample size. To reduce the possible overfitting as much as possible a leave-one-out cross-validation was used. However, future work should be conducted to validate this method with bigger samples. Fifth, cognitive impairment was assessed using the HVLT, and it is assumed that the older adults were healthy. However, they could have some preclinical early pathologic aging that was not controlled such as tauopathy and beta-amyloid accumulation and/or higher atrophy in the hippocampus and frontal regions than the expected for healthy aging. Sixth, within-group differences were not explored, for instance “is it possible to discriminate the individuals with a very bad sleep quality from those with a moderately bad sleep quality?” or “can brain complexity be used to classify individuals with specific sleep disturbances such as latency or number of awakenings per night?” These gaps and questions should be investigated, as they could be the base for creating individualized interventions.

## 5. Conclusions

In conclusion, this study demonstrates that: (i) the algorithm is efficient in classifying older vs. younger participants with good and bad sleep quality; (ii) aging is associated with changes in α θ δ; (iii) bad sleep in older adults is associated with δ changes in the fronto-temporal region while bad sleep both in young and older adults is associated with changes in the occipital region. The algorithm could be used to explore intragroup differences and predict sleep intervention efficiency depending on brain complexity. Future studies using EEG and non-linear features along with ML techniques might enable to predict which intervention is better depending on age, lifestyle, and brain configuration to improve sleep quality [60].

## Figures and Tables

**Figure 1 sensors-24-02811-f001:**
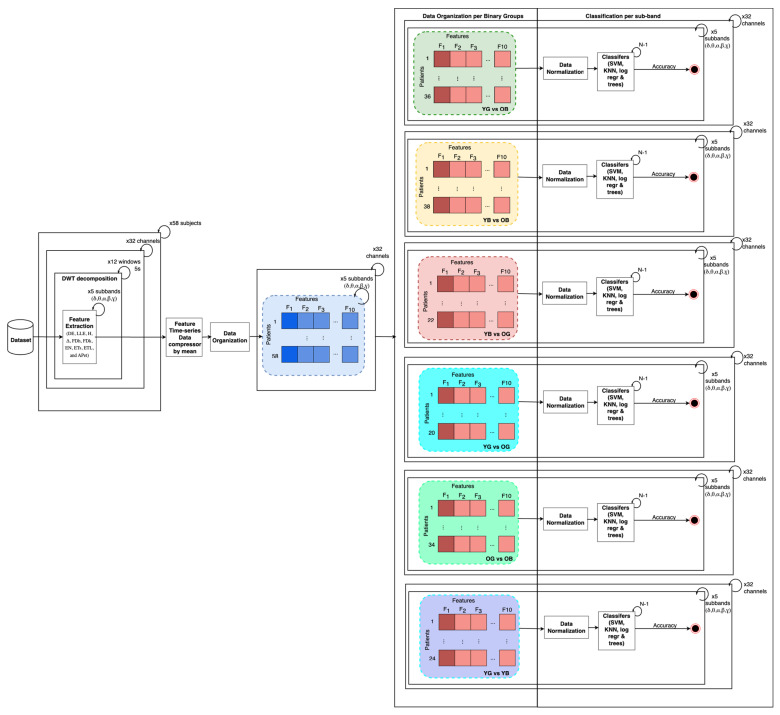
Methodology overview. Left to right, 32 channels included in analysis, time series of 32 channels selected and split into 5 s windows, Discrete Wavelet Transform (DWT) applied to achieve conventional sub-bands per window, electrode and subject. Features extracted for each sub-band, window, and each subject, and then per channel an average of each feature time series vector has been computed. Non-linear features organized per binary groups and sub-band and z-score normalization performed per study group pairs. Non-linear features normalized were input of classifiers trained/tested within a leave-one-out-cross-validation procedure. SVM = Support Vector Machine; KNN = K-Nearest Neighbor; Log.reg = Logistic Regression; trees = decision trees.

**Figure 2 sensors-24-02811-f002:**
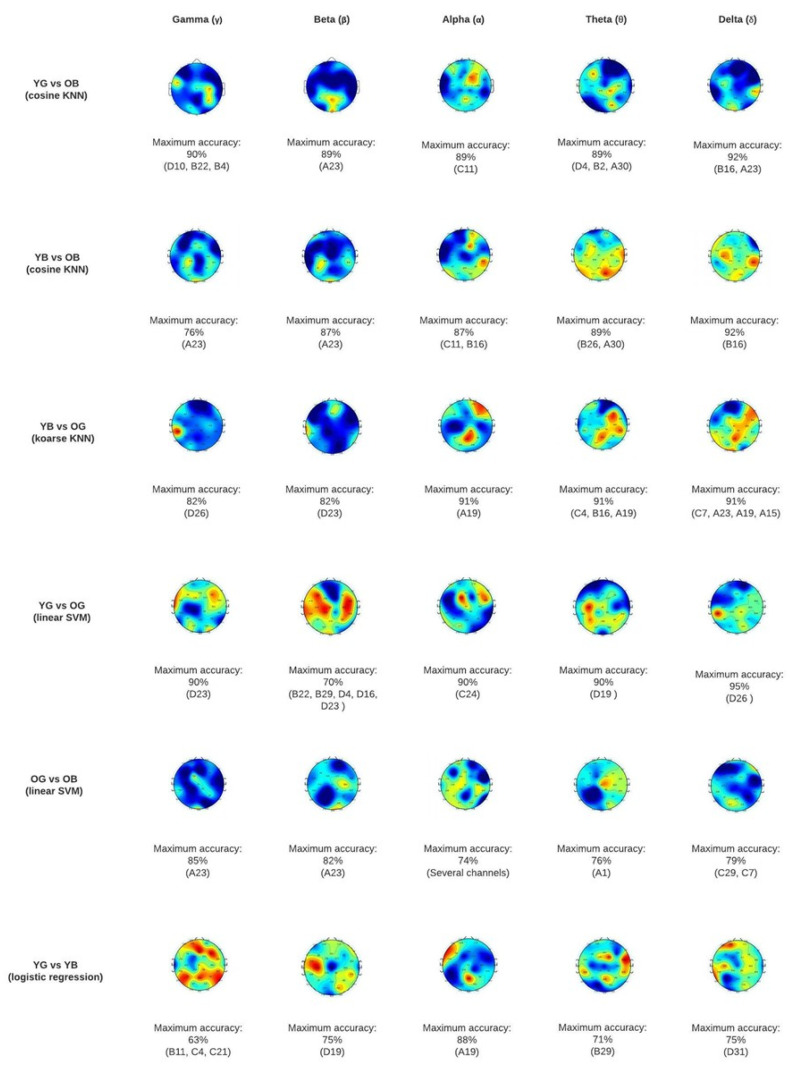
Topographic map classification results at scalp level for each pair of groups. YG = Young adults’ good sleep; YB = Young adults’ bad sleep; OG = Older adults’ good sleep; OB = Older adults’ bad sleep.

**Figure 3 sensors-24-02811-f003:**
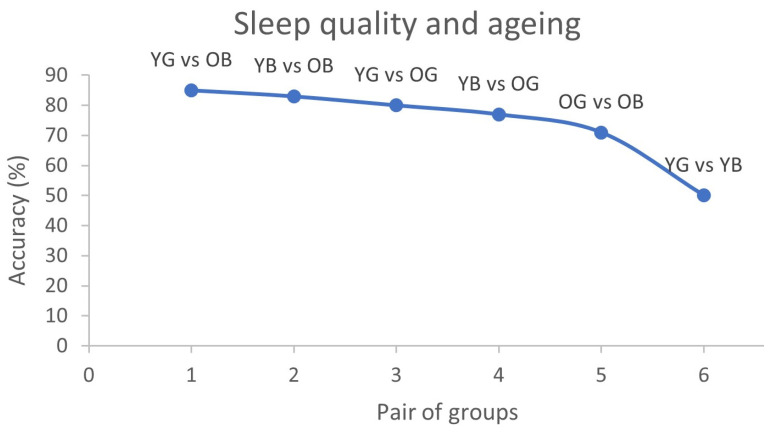
Mean accuracy in each pair of groups that displays how accurately algorithm predicted sleep quality and aging considering brain complexity of each participant. Y axis accuracy percentages are depicted and in X axis six pairs of groups: YG = Young adults’ good sleep; YB = Young adults’ bad sleep; OG = older adults’ good sleep; OB = Older adults’ bad sleep.

**Table 1 sensors-24-02811-t001:** Classifiers and optimized hyperparameters.

Classification Models	Classifier	Optimized Hyper-Parameters
Decision Trees	Fine Tree	Maximum number of splits = 4
	Medium Tree	Maximum number of splits = 20
	Coarse Tree	Maximum number of splits = 100
Logistic Regression		Covariance structure: complete
Support Vector Machines (SVM)	Linear SVM	Box constraint level = 3
	Quadratic SVM	Box constraint level = 3
	Cubic SVM	Box constraint level = 4
	Fine Gaussian	Box constraint level = 3
	Medium Gaussian	Box constraint level = 3
	Coarse Gaussian	Box constraint level = 1
K-Nearest-Neighbors (KNN)	Fine KNN	Number of neighbors = 1
	Medium KNN	Number of neighbors = 10
	Coarse KNN	Number of neighbors = 100
	Cosine KNN	Number of neighbors = 10
	Cubic KNN	Number of neighbors = 10
	Weighted KNN	Number of neighbors = 10

**Table 2 sensors-24-02811-t002:** Accuracy level reached per pair of groups per sub-band.

Group	Classifier	Mean/Max	Sub-Bands				
			Gamma	Beta	Alpha	Theta	Delta
YG vs. OB	Cosine KNN	Mean	70%	73%	80%	82%	**85%**
		Max	75%	89%	89%	89%	**92%**
YB vs. OB	Cosine KNN	Mean	68%	72%	77%	78%	**83%**
		Max	76%	87%	87%	89%	92%
YB vs. OG	Coarse KNN	Mean	60%	62%	75%	79%	**80%**
		Max	82%	82%	91%	91%	91%
YG vs. OG	Linear SVM	Mean	59%	56%	67%	70%	**77%**
		Max	90%	70%	90%	90%	95%
OG vs. OB	Linear SVM	Mean	**72%**	**72%**	**72%**	71%	71%
		Max	85%	82%	74%	76%	79%
YG vs. YB	Logistic regression	Mean	43%	**50%**	49%	47%	50%
		Max	63%	75%	88%	71%	75%

Note. YG = Young adults’ good sleep; YB = Young adults’ bad sleep; OG = older adults’ good sleep; OB = Older adults’ bad sleep; Classifier = Machine learning method that classified with higher accuracy level for each pair of groups; Mean = mean global accuracy level of all channels; Max = maximum accuracy achieved in at least one channel. Light orange mean accuracies ≥ 70; dark orange mean accuracies ≥ 80; light blue maximum accuracy in at least one channel ≥ 70; dark blue maximum accuracy in at least one channel ≥ 80.

## Data Availability

The data that support the findings of this study are available upon request from the corresponding author.

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
