# Peer review of "Classification of Sleep Quality and Aging as a Function of Brain Complexity: A Multiband Non-Linear EEG Analysis"

_sensors, 2024, doi:10.3390/s24092811_

Round 1

Reviewer 1 Report

Comments and Suggestions for Authors

The current study uses an algorithm that applies nonlinear brain complexity features to differentiate waking electroencephalography (EEG) states as a function of age and sleep quality. Ten nonlinear features were extracted from multiband EEG analysis, and fed into three classifiers, followed by leave-one-out cross-validation. The algorithm distinguishes older from younger groups. The authors say it could be used to examine within-group differences in older adults, as well as predict the effectiveness of sleep intervention based on their brain complexity.

However, there are many issues that authors need to clarify before acceptance.

 What are the conditions during the sleep study?

Sleep studies typically monitor brain activity, eye movements, heart rate, and breathing patterns.

A sleep study can be done in specialized centers. During sleep, the EEG monitors the stages of sleep and REM and NREM sleep cycles you go through during the night to identify possible disturbances in your sleep pattern. Sleep studies also measure your eye movements, oxygen levels in your blood (via a sensor), heart rate and breathing, snoring, and body movements.

Do you have the following tests?

Polysomnography:

  In polysomnography, a sleep technician monitors a patient who stays overnight in a specialized clinic. Various functions are measured throughout the night, including eye movements, brain and muscle activity, respiratory effort and airflow, blood oxygen levels, body positioning and movements, snoring, and heart rate.

Multiple Sleep Latency Test:

  Many sleep latency tests measure how quickly someone falls asleep and how quickly they enter REM sleep during a daytime nap. This test is primarily used to diagnose excessive daytime sleepiness that may be due to narcolepsy or an unknown cause (idiopathic hypersomnia).

Continuous positive airway pressure titration:

Continuous positive airway pressure (CPAP) is a common treatment for sleep apnea. When sleep apnea is strongly suspected, a two-night sleep study may be an option. In a two-night study, polysomnography was used to diagnose sleep apnea during the first half of the night, and CPAP titration was performed during the second half of the night.

Home testing for sleep apnea.

What is the stage of sleep in your study? Just in the abstract, you mentioned the end of the dream, but you studied all the frequency subsets. It needs to be described in more detail.

The stages of sleep are generally divided into 2 streams: non-rapid eye movement (NREM) sleep and rapid eye movement (REM) sleep. NREM sleep is classified into 4 stages: Stage 1 (sleepiness), Stage 2 (light sleep), Stage 3 (deep sleep), and Stage 4 (deep slow wave sleep), REM sleep is Stage 5.

Nonlinear dynamics is applied to EEG sleep signals to distinguish different sleep stages. This suggests that subtle variations (microstructural organization) in the EEG signal can be characterized more efficiently by non-linear analysis.

On line 236, the authors explain that only one minute was used to analyze the data, and then in 2.4.1. Multiband EEG Decomposition (page 6) for each participant and channel, the EEG time series is divided into 5 s windows or a total of 12 windows. A global window average (assuming averaged EEG epoch) was then calculated, yielding a 5 s average signal per channel and each participant. Can you explain in more detail?

Why was the EEG window first averaged and then frequency multiband EEG decomposition performed?

The correct procedure is to do a frequency decomposition for each window at the single trail level, and then each trail and window feature should be calculated. In the averaging procedure of the EEG, the non-linear information of the signals is lost. This is a basic rule in nonlinear theory.

Also, the trend is included in the delta frequency, in which non-linear characteristics are not sought since it is linear for relatively short time series such as yours. Each feature must be found for each subrange on each trial, and then the average of the subrange trials can be presented for each participant and its channel (32 channels).

What is the nature of NREM sleep deficits (slow wave activity (SWA) 0.5–3 Hz EEG power)? Most of the approaches assume data stationarity and need relatively large time series to provide reliable results. It seems quite difficult to satisfy these assumptions and requirements for a large part of sleep EEG, which contains a large number of phasic events, such as sleep spindles, K-complexes, spike waves, brief awakenings, etc. These phasic events during NREM have a particular arrangement, described as a “cyclic alternating pattern”, which consists of transient complexes of phase A arousal that periodically interrupts the tonic theta delta activities of phase B NREM sleep.

Only subtle variations in the EEG signal can be characterized more efficiently by non-linear analysis. The choice of epochs for the evaluation of sleep EEG dynamics should mainly take into account this microstructural characteristic, mostly at high frequencies.

The following lines have gaps in the description:

line 331: defines Tm as what the average period means.

line 334: Why look for Detrended fluctuation analysis when the delta sub-band has (line 262: δ, 0.1-4 Hz)? Detrended fluctuation analysis quantifies the fine details of physiological signals using the fractal property.

line 351: defines R and S.

line 475: Check the frontotemporal area D23, this is the posterior part of the superior temporal area (classical T7).

line 480: verify that C11 is not frontotemporal, it is the precentral (or frontal) region.

line 491: checks D19 (parietal), this is the motor area (classical C3).

line B29: (frontocentral) is also a questionable position.

line 554: in discussion "Although one minute of EEG data is sufficient for this type of analysis, future studies should include longer time series or assess whether accuracy levels change over different time windows". Many studies show that the results depend on the length of the time window. As with the different time windows, other information about sleep phases is covered.

Finally, in the discussion, the authors should relate the result to sleep quality characteristics.

Reviewer 2 Report

Comments and Suggestions for Authors

(1)The contribution should be clearly stated in the introduction section. The differences of your work and the existing must be analysed.

(2)What algorithm is the proposed in this paper? The used method in this paper are all of conventional machine learning method?  The core theory are not scarce in the paper. 

(3)In Table 1, what is the physical meaning of Gamma, Beta, Alpha, Theta, Delta?

(4)How is to define good sleep and bad sleep? 

(5)The recent references are less. 

(6)The future research plan should be given in the conclusion section. 

Reviewer 3 Report

Comments and Suggestions for Authors

Any specific Reason for selecting 58 subjects?

Can you specify any special reasons for 24 young adult,  sleepers11 and 34 older adults?

What is the difference you feel in between Lower accuracy and higher accuracy?

Introduction part was too long to read.

Fifty-eight right-handed volunteers were included in the current study. DId you tried with left handed?

Any specific reason for Selecting MATLAB ver. R2018a? because latest version available now.

Why did you select DWT for your study? latest feature extraction methods are available now.

can you explain Detrended Fluctuation Analysis in detail?

Accuracy of the study was not sufficent.

Result and discussion was not sufficient.

So i kindly request the authors to improve the paper 

Round 2

Reviewer 2 Report

Comments and Suggestions for Authors

The authors have completed the paper revisions according to the reviewers' comments.